# Feasibility of EEG Phase-Amplitude Coupling to Stratify Encephalopathy Severity in Neonatal HIE Using Short Time Window

**DOI:** 10.3390/brainsci12070854

**Published:** 2022-06-29

**Authors:** Xinlong Wang, Hanli Liu, Eric B. Ortigoza, Srinivas Kota, Yulun Liu, Rong Zhang, Lina F. Chalak

**Affiliations:** 1Department of Bioengineering, University of Texas at Arlington, Arlington, TX 75220, USA; xl.victor.wang@gmail.com (X.W.); hanli@uta.edu (H.L.); 2Division of Neonatal-Perinatal Medicine, Department of Pediatrics, University of Texas Southwestern Medical Center, Dallas, TX 75220, USA; eric.ortigoza@utsouthwestern.edu; 3Department of Neurosurgery, University of Texas Southwestern Medical Center, Dallas, TX 75220, USA; svkota@gmail.com; 4Department of Population and Data Sciences, University of Texas Southwestern Medical Center, Dallas, TX 75220, USA; yulun.liu@utsouthwestern.edu; 5Departments of Internal Medicine and Neurology, University of Texas Southwestern Medical Center, Dallas, TX 75220, USA; rongzhang@texashealth.org

**Keywords:** neonatal hypoxic-ischemic encephalopathy, time-dependent phase amplitude coupling, EEG, hypoxic ischemia encephalopathy biomarkers, mixed effects models

## Abstract

Goal: It is challenging to clinically discern the severity of neonatal hypoxic ischemic encephalopathy (HIE) within hours after birth in time for therapeutic decision-making for hypothermia. The goal of this study was to determine the shortest duration of the EEG based PAC index to provide real-time guidance for clinical decision-making for neonates with HIE. Methods: Neonates were recruited from a single-center Level III NICU between 2017 and 2019. A time-dependent, PAC-frequency-averaged index, tPAC_m_, was calculated to characterize intrinsic coupling between the amplitudes of 12–30 Hz and the phases of 1–2 Hz oscillation from 6-h EEG data at electrode P3 during the first day of life, using different sizes of moving windows including 10 s, 20 s, 1 min, 2 min, 5 min, 10 min, 20 min, 30 min, 60 min, and 120 min. Time-dependent receiver operating characteristic (ROC) curves were generated to examine the performance of the accurate window tPAC_m_ as a neurophysiologic biomarker. Results: A total of 33 neonates (mild-HIE, *n* = 15 and moderate/severe HIE, *n* = 18) were enrolled. Mixed effects models demonstrated that tPAC_m_ between the two groups was significantly different with window time segments of 3–120 min. By observing the estimates of group differences in tPAC_m_ across different window sizes, we found 20 min was the shortest window size to optimally distinguish the two groups (*p* < 0.001). Time-varying ROC showed significant average area-under-the-curve of 0.82. Conclusions: We demonstrated the feasibility of using tPAC_m_ with a 20 min EEG time window to differentiate the severity of HIE and facilitate earlier diagnosis and treatment initiation.

## 1. Introduction

Neonatal hypoxic-ischemic encephalopathy (HIE) remains a major cause of morbidity and mortality in both developing and developed countries. HIE presents clinically with a dynamic and fluctuating course, which is often difficult to classify immediately after birth and can jeopardize the timely initiation of therapeutic interventions to mitigate injury. Randomized controlled trials (RCT) of infants with moderate to severe HIE have clearly established that therapeutic hypothermia (TH) significantly reduces death or disability by 25% when started within 6 h of birth (number needed to treat is 6) [1,2,3]. The difficulty of early and accurate diagnosis of the severity of neurologic injury is complicated by the natural evolution of the insult and the clinical overlap between mild and moderate HIE. Given the lack of sensitive and timely biomarkers, the clinical decision-making regarding initiation of TH can be challenging, and neonates who need treatment can go unrecognized. There is an urgent clinical need for improved diagnostic tools to classify the severity of neonatal HIE immediately after birth. An ideal biomarker of HIE would be measured in real time and directly reflect the neurovascular unit function linking it to outcomes. Such a biomarker would enhance the ability to stratify the insult severity by identifying neonates who might benefit from hypothermia. The quest for such biomarkers is still mostly research based. While brain neuroimaging with MRI/MRS with lactate/NAA ratio in the thalamus or fractional anisotropy in the PLIC [4,5,6] before discharge is the gold standard to predict long-term neurodevelopment [7], it offers one snapshot in time as it is not available for dynamic real time measures or in the first day of life for decision making to enroll in trials. Recently, our group has reported the EEG based phase amplitude coupling (PAC) between slow and fast brain oscillations as a novel EEG marker of neural synchrony, which becomes dysregulated after injury.

The human brain is a unified entity that works as a comprehensive system rather than single parts [8,9]. The successful coordination between neural oscillators requires accurate synchronizations and efficient communications, which facilitate one of the key features of brain oscillations, cross-frequency phase amplitude coupling (PAC) [10,11,12]. PAC usually refers to the phase of a slow frequency brain oscillation modulating upon the amplitude of a higher frequency brain oscillation measured by intracortical/scalp EEG, which has been reported as a general mechanism mediating the encoding, storage, and retrieval of information [13,14,15,16]. Therefore, the PAC index can potentially serve as a biomarker to describe the efficiency of the neural information process in developed human brains [9,17].

The neonatal EEG is distinct from the EEG recorded in later infancy. Specifically, EEG signals from premature neonates have particular hallmarks called “Delta brushes”, which refer to the transient patterns comprising a slow delta wave (0.5–2 Hz) and superimposed fast activity (8–30 Hz) [18,19], reflecting/indicating the maturity of the neonatal brain development. Since the occurrences of Delta brushes are prominent and abundant in normal neonatal EEG, the abnormalities in Delta brushes are diagnostically and prognostically useful in the recognition of suppressed electrophysiology [20]. PAC analysis has been validated as an effective method to quantify the age-related occurrence of the Delta brushes and monitor brain maturity [19]. Thus, PAC in the neonatal brains indicates intact cerebral functions with neural oscillation synchrony and the strength of PAC can be used to evaluate disruptions in cortical function caused by HIE. In our prior investigation, we showed that the mean index of phase amplitude coupling (PAC_m_) obtained from a 6-h EEG recording can identify the severity of neonatal HIE needing TH [21].

However, a real-time metric with a shorter data acquisition window is necessary to improve the diagnostic timeliness for initiation of TH within the therapeutic window. Therefore, the objective of this study was to build upon our prior work by testing a short time-dependent PAC_m_ (tPAC_m_) to improve the timeliness of diagnosing HIE severity.

## 2. Methods

### 2.1. Participants

This study was approved by the institutional review board at the University of Texas Southwestern Medical Center Institutional Review Board at an initial approval data of 23 June 2015. The project identification code was STU 022015-104, and the full title of the study was: A Novel Approach to Quantification of Cerebrovascular Function in Newborns.

A parent of each neonate signed a written informed consent prior to enrollment. All recruited newborns were born at 36 weeks gestational age or greater, weighed more than 1800 g at birth, had evidence of metabolic acidosis, with signs of encephalopathy within the first six hours of life (HOL) along with a sentinel perinatal event. Infants were classified according to the modified Sarnat staging from the National Institute of Child Health and Human Development (NICHD) and neonates were classified with moderate or severe HIE received TH. All recordings were obtained prior to initiation of cooling in the first 6 h of life.

The NICHD (National Institute of Child Health and Human Development) inclusion criteria at <6 HOL (Hours of Life) is used to screen infants for eligibility see Table A1 for more details.

HIE_mild_/No encephalopathy: Fetal Acidosis with an acute labor complication such as stat section, placental abruption, meconium (perinatal acidosis is defined by NICHD40, with pH < 7.15 in a cord gas or the first blood gas available or base deficit ≥ 10 mmol/L).

HIE_cooled_: Fetal acidosis as above with additional presence of encephalopathy needing cooling with three of six categories in the modified Sarnat exam showing moderate to severe abnormalities (level of consciousness, posture, tone, moro and suck reflexes, and autonomic system breathing). Detailed categories are presented in the Appendix A Table A1.

Exclusion Criteria included: genetic syndromes; birthweight < 1800 g; and/or head circumference < 30 cm as those can interfere with the primary outcome.

For analysis, newborns were grouped in two major categories: fetal acidosis with mild encephalopathy (HIE_mild_) and those with moderate to severe (according to the modified Sarnat exam) HIE requiring treatment with TH (HIE_cooled_). Whole-body hypothermia treatment was conducted on the neonates in the group of HIE to maintain a core temperature of 33.5 °C for 72 h, followed by rewarming at 0.5 °C per 1–2 h using a servo-controlled blanket (Blanketrol II, Cincinnati Sub-Zero, Cincinnati, OH, USA) according to the NICHD protocol. For enrolled infants, neuromonitoring was initiated within the first six hours of birth (average at 4 h) and continued for a duration of 24 h.

### 2.2. EEG Data Acquisition and Pre-Processing

The EEG data acquisition was conducted at a sampling frequency of 256 Hz and at eight different locations, including C3, C4, Cz, Fz, O1, O2, P3, and P4, based on the standard 10–20 montage modified for newborns (Nihon Kohden, Irvine, CA, USA). A Moberg Component Neuromonitoring System monitor (Moberg Research, Inc., Ambler, PA, USA) allowed same scalp EEG signals with other physiological parameters to be recorded. The investigated parts of the EEGs were seizure-free according to visual inspection by a neonatologist.

As published prior [21], the same first 6-h EEG signals were processed offline using MATLAB (MathWorks Inc., Natick, MA, USA). First, the 4th order Butterworth filters were applied to filter the EEG signals with a high-pass cutoff frequency of 0.5 Hz and a low-pass frequency of 70 Hz using the native Matlab function ‘filtfilt’, which was to minimize the effects of frequency-dependent phase shifts caused by the filters. Next, the 60 Hz power line noise and the interference frequency with other recording systems were further removed using notch filters at 60 Hz and 47 Hz. Moreover, a re-referencing procedure was conducted on each EEG electrode to the common mean of the eight electrodes.

Finally, the motion artifacts in the EEG signals were carefully identified and removed. Specifically, the EEG signals were segmented into 1 s epochs; the signal standard deviation in each epoch was calculated. If this value was larger than 50 μV or less than 3 μV, this epoch was considered contaminated by high motion artifacts or bad electrode-tissue contact and was excluded from further processing. This procedure removed the segments of EEG signals with low signal-to-noise-ratio, such as the periods with “huge spikes” or “flat lines/recordings”. After these, for each electrode, sampling outliers were identified and removed if their scales (the absolute value of each EEG sample) were larger than four times the standard deviation of the temporal mean (all EEG samples across the full length, i.e., all the epochs) in each electrode. After all the pre-processing procedures above, the original 6 h EEG data were shortened due to the removal of noisy/artifact-affected epochs, resulting in different temporal lengths of data for each subject. Subjects with artifact-free EEG readings less than 30 min duration were excluded from further data analysis. Two sample *t*-tests were conducted to confirm that there were no statistical differences in the total time lengths of artifact-free EEG data between HIE_mild_ and HIE_cooled_ groups.

### 2.3. Time-Dependent PAC_m_ Quantification

The major steps of PAC_m_ quantification in any EEG time window is demonstrated below: (1) one broadband EEG time series was 1–2 Hz bandpass filtered as the slow oscillation component in PAC; then; (2) the same broadband EEG time series was 12–30 Hz bandpass filtered as the fast oscillation component in PAC; Next; (3) Hilbert Transform was utilized to obtain the time-dependent phase alteration of the slow oscillation, *θs(t)*, and the amplitude envelop of the fast oscillation, *Af(t)*. Finally, the calculation of *tPAC* can be conducted as Equation (1):(1)$$tPAC_{m}=\frac{1}{\sqrt{n}}\times \left|\frac{\sum_{t=1}^{n}A_{f}\left(t\right)e^{i\theta_{s}\left(t\right)}}{\sqrt{\sum_{t=1}^{n}A_{f}{\left(t\right)}^{2}}}\right|$$
where *n* denotes the number of EEG samples (data points) in an EEG time series, *θs(t)* denotes the time-dependent phase of 1–2 Hz bandpass filtered EEG data, while *Af(t)* denotes the 12–30 Hz bandpass filtered EEG from the same time series.

Moving-window based time-dependent PAC_m_ indexes (tPAC_m_) were quantified with different window sizes and 50% sliding window overlap. In this study, instead of calculating one single PAC index for the whole 6-h EEG signal, a series of tPAC_m_ indexes were calculated based on the moving window method. The window sizes included 10 s, 20 s, 1 min, 2 min, 5 min, 10 min, 20 min, 30 min, 60 min, and 120 min. The PAC quantification procedures were consistent with our previous work [21]. Briefly, direct mean vector length (MVL) was performed to obtain the coupling between the amplitude of 12–30 Hz waves versus the phase of 1–2 Hz waves.

Figure 1 depicts a moving-window-based tPAC_m_ calculation using 20 min window size and EEG data at the P3 EEG electrode as an example. The solid red window denotes the first 20 min window, which is used to quantify the first tPAC_m_ index of the EEG recording. Next, this window is moved forward with a step increment of 10 min (i.e., 50% of the window size/length) until the end of the EEG, resulting in a series of tPAC_m_ indexes.

Following the abovementioned calculation procedures, tPAC_m_ indexes in all the 10 different window lengths and at all the 8 EEG electrodes were quantified. Our earlier work reported that PAC_m_ indexes at electrode P3 were the most sensitive to differentiate newborns with moderate/severe HIE versus mild HIE when using the whole 6 h length of data in the quantification. Therefore, time dependent tPAC_m_ curves at electrode P3 are plotted in Figure 2 as a demonstrative result, while tPAC_m_ curves from other electrodes are shown in Appendix A Figure A2, Figure A3, Figure A4, Figure A5, Figure A6, Figure A7 and Figure A8.

### 2.4. Statistical Analysis for Repeated Measures Using the Linear Mixed-Effects Models

In this study, repeated measures of tPAC_m_ were quantified by sliding-window-based PAC calculations and were compared between mild HIE versus moderate/severe HIE neonates. To evaluate associations between repeated measures of tPAC_m_ and study groups (HIE_mild_ versus HIE_cooled_), linear mixed-effects models were performed with an unstructured covariance matrix and the maximum likelihood estimation method.

The linear mixed-effects model [22], which is an extension of linear regression models, allows us to model tPAC_m_ data that are correlated and measured repeatedly over time by using both fixed and random effects. Specifically, the effect of the study group effect was modelled as a fixed effect because we expected that there would be an average relationship between study group and tPAC_m_, while participants were modelled as random effects to account for variability across subjects.

The models included study group (HIE_mild_ versus HIE_cooled_), time (treated as a continuous variable), and their interaction term (i.e., group × time) as fixed effects, and a random effect for neonates to account for within-subject correlation of repeated measures over time. The above analyses were conducted repeatedly for all the window sizes respectively using the R software (Vienna University of Economics and Business, Wien, Austria), version 4.0.3. All statistical tests were two-sided, and *p* < 0.05 was considered as significant.

### 2.5. Statistical ROC Analysis

Using 20 min window size on electrode P3, time-dependent classification performance of the tPAC_m_ index on differentiating HIE_mild_ versus HIE_cooled_ neonates was quantified by each individual ROC curve at each temporal point. These ROC curves are plotted in Figure 3b.

## 3. Results

A total of 33 term newborns admitted to the Parkland Health and Hospital System neonatal intensive care unit (NICU) from November 2017 to December 2019 were recruited for our prior study and utilized in this novel analysis. Detailed clinical information is summarized in Table 1.

### 3.1. Determination of Time-Dependent tPAC_m_ from Both HIE Neonate Groups

Dynamic tPAC_m_ values at electrode P3 from mild-HIE (blue) and moderate/severe HIE (red) are plotted in Figure 2a–j, respectively for each window size. The shaded error bar indicates the standard error of the mean. The tPAC_m_ values at each of the remaining electrodes are included in the Appendix A. Specifically, Figure A2a is for window size of 10 s; (b) is for 20 s; (c) is for 1 min; (d) is for 2 min; (e) is for 5 min; (f) is for 10 min; (g) is for 20 min; (h) is for 30 min; (i) is for 1 h; (j) is for 2 h. An increasing trend of separation between the two groups was observed as the window size increased.

**Figure 2 brainsci-12-00854-f002:**
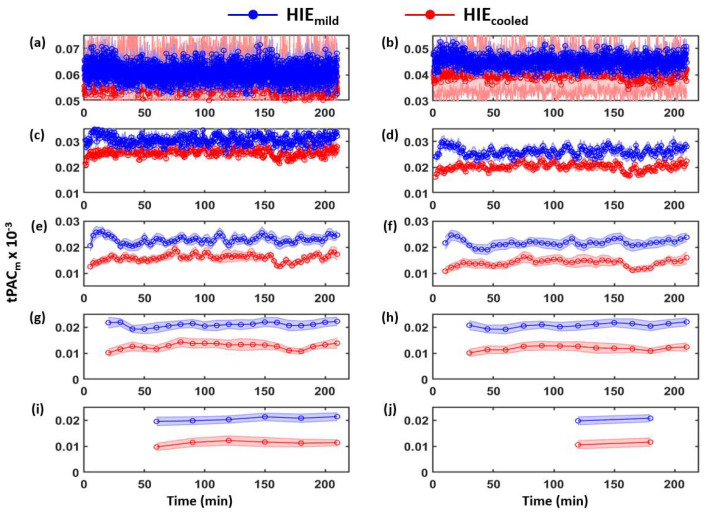
tPAC_m_ indexes at P3 from 15 HIE_mild_ (blue) and 18 HIE_cooled_ (red) under (**a**) 10 s; (**b**) 20 s; (**c**) 1 min; (**d**) 2 min; (**e**) 5 min; (**f**) 10 min; (**g**) 20 min; (**h**) 30 min; (**i**) 60 min; (**j**) 120 min window sizes. Each point denotes the cross-neonate average of tPAC_m_ for the group at different times. The shaded error bar indicates the standard error of the mean. t = 0 indicates the start of EEG recording. HIE infants meeting cooling criteria are shown in red. Infants with mild or no encephalopathy by hospital discharge are shown in blue. HIE: Hypoxic Ischemic Encephalopathy.

Results of linear mixed-effect models for electrodes P3 are presented in Table 2. For the 10 s window size, the mixed-model estimated difference in tPAC_m_ values between the two study groups was −3.62 (95% CI = (−6.00, −1.25); *p* = 0.04); For the 20 s window size, the mixed-model estimated difference in tPAC_m_ values between the two study groups was −3.90 (95% CI = (−6.16, −1.64); *p* = 0.001), noted by the significant main effects of time (*p* < 0.001) for these 10 s and 20 s windows while still able to identify significantly lower tPAC_m_ (×10^−3^) in HIE.

Importantly, neither the main effect for time nor the group × time interaction was significant for tPAC_m_ values for all window sizes over 1 min (Table 2).

In summary, tPAC_m_ (×10^−3^) quantified with >1 min window length was significantly lower in the HIE_mild_ group than in the HIE_cooled_ group without time-dependent effects.

In Figure 3a, the relationship of the coefficient estimate group (HIE_mild_ vs. HIE_cooled_) shown in Table 2 and their window sizes were plotted. A decreasing trend of coefficient estimate was observed as window size increased. Of note a plateau in the coefficient estimate was observed with window size of 20 min or more.

**Figure 3 brainsci-12-00854-f003:**
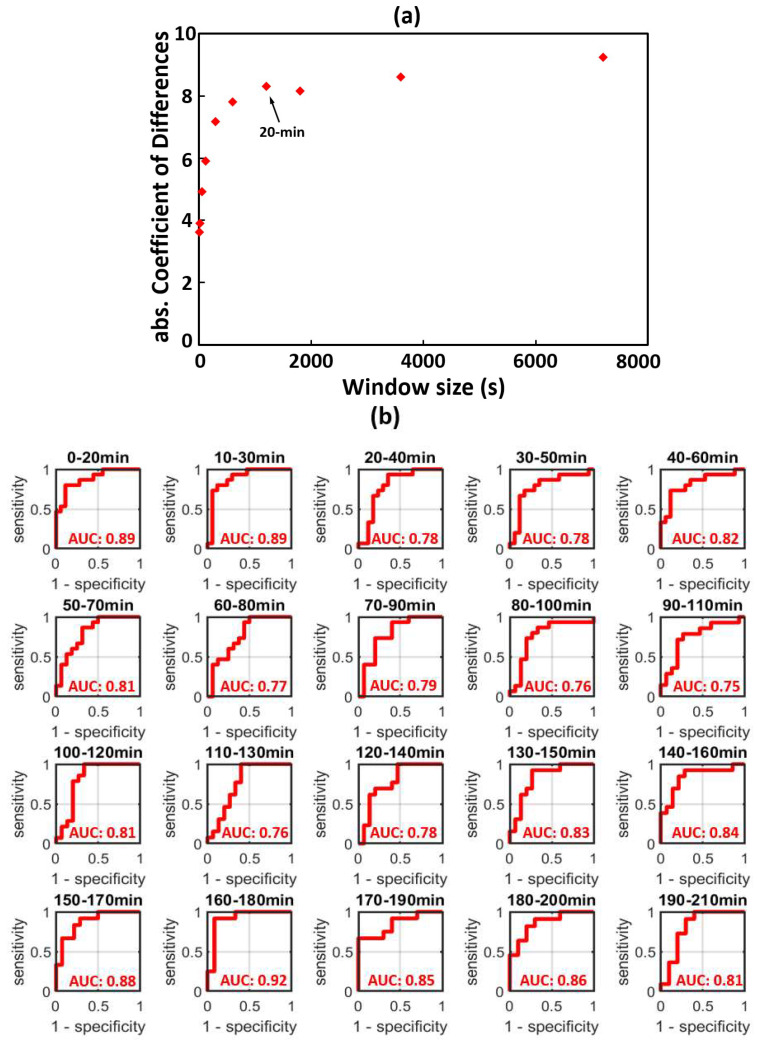
(**a**) Relationship of window sizes versus the absolute values of coefficient estimates for all 10 different windows. The red points denote the coefficients estimates for different window sizes in Table 2. The arrow indicates the shortest optimal window, 20 min, to separate the HIE_mild_ and HIE_cooled_ groups; (**b**) using 20-min window tPAC_m_, the time-dependent Area Under Curve (AUC) to classify the HIE_mild_ neonates versus HIE_cooled_, at electrode P3. AUC: Area Under Curve.

Therefore, 20 min window size was selected as the shortest window to optimize the separation of tPAC_m_ indexes between HIE_mild_ and HIE_cooled_.

### 3.2. Time-Dependent ROC Classification Using 20 min Window tPAC_m_ to Differentiate between HIE_mild_ and HIE_cooled_

Based on the results demonstrated above, a 20 min window size was selected as the optimal window to quantify tPAC_m_ and separate HIE_mild_ and HIE_cooled_. Time-dependent ROC curves were generated at each 20 min time window to quantify the performances of tPAC_m_ in classifying HIE_mild_ versus HIE_cooled_ (Figure 3b) at electrode P3. The time-dependent AUC values were also quantified and noted in each respective ROC graph. A temporally averaged AUC of 0.82 (±0.05) was obtained.

## 4. Discussion

In this study, we build upon our prior work demonstrating the utility of EEG PAC as a biomarker to classify the severity of neonatal HIE. This EEG based physiological biomarker was promising for early objective classification of HIE yet required 6 h of recording [21]. We aimed in this concept paper to improve the PAC timeliness needed for meaningful clinical decision-making. We developed and tested a time-dependent, PAC-frequency-averaged index (tPAC_m_) to characterize intrinsic coupling between the amplitudes of 12–30 Hz EEG rhythms and the phases of 1–2 Hz oscillations using a shorter time window that would allow timely diagnosis of the severity of neonatal HIE. We demonstrated that the tPAC_m_ 20 min time window provided a robust neurophysiologic biomarker with a temporally averaged AUC of 0.82 for discrimination between HIE_mild_ and HIE_cooled_. In a disease process characterized by a fluctuating course with poor diagnostic tools, this methodology has the potential to transform neonatal HIE care models by facilitating earlier identification of HIE severity and prompt initiation of treatment.

The strengths of this study include the use of a well-described clinical cohort of neonates with HIE as well as a robust analytic approach of different sizes of moving time windows, including 10 s, 20 s, 1 min, 2 min, 5 min, 10 min, 20 min, 30 min, 60 min, and 120 min. As demonstrated in Figure 3a, using a mixed-effects statistical model, we found that the averaged coefficient estimates between HIE_mild_ and HIE_cooled_ were enhanced as the window size increased. Specifically, such differences between the two groups became larger and clearer as the time window lengthened from 20 to 120 min while they were visually inseparable when the window was 10–20 s long. Notably, an inflection point was observed in Figure 3a at the 20 min window mark, showing an optimal window length that permits accurate distinction in tPAC_m_ between the two groups.

The current clinical need is to develop neuromonitoring tools that can bolster clinical exam and laboratory data to determine HIE severity promptly and accurately in neonates at the bedside, in order to facilitate timely initiation of neuroprotective therapies such as TH [1,23,24,25]. This requires that the tPAC_m_ calculation be accurate while requiring a short time period of patient data acquisition. Therefore, the selection of optimal window length of tPAC_m_ calculation is based on two criteria, (1) the window length should provide distinct tPAC_m_ indexes to optimally separate HIE_mild_ versus HIE_cooled_, and (2) the window length should be as short as possible to facilitate timely clinical decision making. Table 2 shows that tPAC_m_ values were statistically distinct between the two neonatal groups without any interaction with time when the window length was equal to or longer than 1 min.

This implies a minimum EEG data acquisition time (with a sampling frequency of 256 Hz) to be at least 1 min for accurate differentiation between HIE_mild_ versus HIE_cooled_. In other words, significant separation in tPAC_m_ indexes between the two groups of neonates was dependent on the time factor when a 10 s or 20 s window was chosen. This may result from inadequate EEG sample points during this short period of time, as evidenced by a poorly formed histogram for PAC_m_ index calculation (see Figure A1 for more details). Inadequate data would create large uncertainty and variation in determining tPAC_m_ indexes at each time point [13,26]. When the window size gradually increased from 1 min to 20 min, more EEG data points were included in the calculation of tPAC_m_ and thus gave rise to smoother temporal tPAC_m_ traces because most of the noisy oscillation elements were temporally averaged out. In summary, the involvement of larger time windows facilitates more temporally stable tPAC_m_ indexes, while sacrificing the temporal or timely promptness for tPAC_m_ onsite display. In this study, the loss of timeliness was minimized by optimally selecting 20 min as the shortest and optimized window length to sustain sufficient distinction between HIE_mild_ versus HIE_cooled_.

As illustrated above, the main clinical advantage of tPAC_m_ is its flexibility of using shorter time windows to identify HIE severity. This can assist clinicians in making timely decisions regarding use of neuroprotective therapies in neonates with HIE. To the best of our knowledge, using time-dependent PAC for HIE severity identification is a novel approach in this field. Others have employed the power of conventional EEG and the traces of amplitude EEG to identify HIE [27,28,29,30,31], but the use of tPAC_m_ had not yet been explored. Moreover, our group has recently implemented neurovascular coupling to identify impaired cerebral autoregulation in neonates with HIE utilizing the wavelet coherence between clinical SO_2_ and aEEG recordings [32,33]. Although these methods have shown promising results in differentiating severity of HIE, they all require a long recording time of at least 6 h to make accurate clinical decisions, which limits their utility in the modern TH era. Therefore, the tPAC_m_ approach with a shorter time window reported in this study can serve as an additional measure on top of the existing tools to facilitate timely identification of HIE severity at the bedside, since the clinical essence of HIE is to offer continuous monitoring, so as to classify the persistent grade of encephalopathy at discharge. The observations in this study implied that averaging windows of 10–20 min can be adapted for algorithms in the future for continuous HIE monitoring that can guide decision making regarding hypothermia in real time.

On the other hand, several limitations to this study necessitate further work to develop tPAC_m_ as a biomarker of HIE severity. One limitation of this study is that the tPAC_m_ data in some scenarios slightly violated the assumption of normality when conducting mixed-effects models. Thus, caution in interpretation of the results from this study is necessary. In addition, PAC_m_ calculation might be affected by the gaps between EEG epochs after artifact removal. Nevertheless, the removed lengths of EEG data among neonates were identical between the two groups (*p* > 0.05 in two sample *t*-tests). So, there were similar numbers of “gaps” in the EEG data between the two groups. If there was impact on PAC, the effects would be similar. Moreover, the number of patients included in the study demonstrates feasibility, but validation of tPAC_m_ to diagnose severity of HIE in larger cohorts is necessary. Similarly, due to the small sample size, we were not able to assess differences based on gestational age, although neural synchrony increases with age rapidly as the fetus nears term gestation and in the first weeks of postnatal life. In larger cohorts, gestational age-specific cutoffs may be needed, particularly as TH and adjuvant neuroprotective therapies are being increasingly applied to preterm populations. Likewise, in this study we found the most accurate differentiation of tPAC_m_ between the HIE_mild_ and HIE_cooled_ groups at the P3 EEG electrode. This may be unique to our study population and therefore requires further exploration in larger and more diverse cohorts of neonates with HIE. Finally, a critical step to bring this technology to the clinical realm will be the development of bedside systems that can perform the complex analyses required to interpret these dynamic signals, which are prone to artifact and currently not automated. The methodology of tPAC_m_ quantification currently requires an offline preprocessing step for artifact removal. In the future, the preprocessing procedure will be incorporated and automated into a user-friendly graphical user interface to facilitate the onsite display and prompt interpretation of tPAC_m_ results by clinicians.

## 5. Conclusions

The time-dependent, PAC-frequency-averaged index (tPAC_m_) quantified using 20 min EEG time windows is a sensitive biomarker to differentiate severity of HIE, and with further development, has the potential to facilitate earlier diagnosis and treatment initiation for neonates with HIE.

## Figures and Tables

**Figure 1 brainsci-12-00854-f001:**
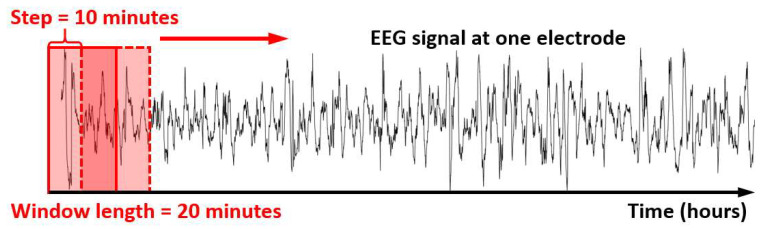
Demonstration of the sliding window EEG quantification of dynamic tPAC_m_ using 20 min window size and EEG data at P3 electrode as an example. The solid red box denotes the first 20 min window used to calculate the first tPAC_m_ index. The dashed box denotes the next sliding-window selection of EEG signals based on a 10 min step (50% of the window length), which is used to calculate the second tPAC_m_ index. The window moves across the total length of EEG signal till the end, producing a series of tPAC_m_ indexes time-dependently. EEG: Electroencephalogram.

**Table 1 brainsci-12-00854-t001:** Characteristics of the neonatal cohort.

Neonatal Characteristics	Overall	Encephalopathy Grade	
		HIE_mild_	HIE_cooled_
Total *N*	33	15	18
Male: *N* (%)	19 (58%)	10 (67%)	9 (50%)
Gestational Age (weeks), mean (SD)	39 (1.3)	39 (1.1)	39 (1.4)
Birth Weight (kg), mean (SD)	3.3 (0.7)	3.3 (0.5)	3.3 (0.8)
Apgar 1 min *, median (IQR)	2 (1 3)	3 (2 4)	1 (1 2)
Apgar 5 min *, median (IQR)	6 (4 7)	7 (6 8)	4 (2 6)
Umbilical Cord Gas pH, mean (SD)	7.0 (0.1)	7.0 (0.1)	7.0 (0.2)
Base Deficit, mean (SD)	16.6 (6.2)	17.6 (3.8)	15.6 (7.6)
Maternal Race/Ethnicity: *N* (%)			
Caucasian non-Hispanic	2 (6%)	1 (7%)	1 (6%)
Black non-Hispanic	8 (24%)	4 (27%)	4 (22%)
Hispanic	21 (64%)	9 (60%)	12 (67%)
Other non-Hispanic	2 (6%)	1 (7%)	1 (6%)
Delivery Mode: *N* (%)			
Caesarean	20 (61%)	8 (53%)	12 (67%)
Vaginal	13 (39%)	7 (47%)	6 (33%)
Maternal Risk Factors: *N* (%)			
Hypertension	8 (24%)	4 (27%)	4 (22%)
Diabetes	2 (6%)	1 (7%)	1 (6%)
Pre-eclampsia	9 (27%)	3 (20%)	6 (33%)
Labor Complications: *N* (%)			
Meconium	9 (27%)	2 (13%)	7 (39%)
Placental Abruption	2 (6%)	1 (7%)	1 (6%)
Uterine Rupture	2 (6%)	1 (7%)	1 (6%)
Maternal Chorioamnionitis	9 (27%)	5 (33%)	4 (22%)
Placental Chorioamnionitis	19 (58%)	9 (60%)	10 (56%)
Disposition:			
DOL at discharge *, median (IQR)	9 (6 16)	6 (5 7)	14 (9 20)
Death prior to discharge	1 (3%)	0 (0%)	1 (6%)

* indicates significance with *p* < 0.05; DOL: Days of Life. SD: standard deviation; IQR: interquartile range; HIE: hypoxic ischemic encephalopathy.

**Table 2 brainsci-12-00854-t002:** Mixed-effects models for different window sizes on P3.

Mixed Effect Models			
Time-Window	Variable	Coefficient Estimates (95% CI)	*p*-Value
10 s	Group (HIE_mild_ vs. HIE_cooled_)	−3.62465 (−6.00060, −1.24869)	0.004 *
	Time	−0.00043 (−0.00059, −0.00027)	<0.001 *
	Group × Time	−0.00006 (−0.00029, 0.00016)	0.594
20 s	Group (HIE_mild_ vs. HIE_cooled_)	−3.90037 (−6.16340, −1.63734)	0.001 *
	Time	−0.00063 (−0.00099, −0.00028)	<0.001 *
	Group × Time	−0.00005 (−0.00055, 0.00045)	0.842
1 min	Group (HIE_mild_ vs. HIE_cooled_)	−4.92412 (−5.42233, −4.42591)	<0.001 *
	Time	−0.00013 (−0.00170, 0.00143)	0.868
	Group × Time	0.00051 (−0.00166, 0.00269)	0.644
2 min	Group (HIE_mild_ vs. HIE_cooled_)	−5.91112 (−8.69999, −3.12225)	<0.001 *
	Time	−0.00148 (−0.00512, 0.00216)	0.425
	Group × Time	−0.00154 (−0.00666, 0.00358)	0.556
5 min	Group (HIE_mild_ vs. HIE_cooled_)	−7.17580 (−10.44403, −3.90756)	<0.001 *
	Time	−0.00362 (−0.01689, 0.00964)	0.592
	Group × Time	−0.00089 (−0.01953, 0.01776)	0.926
10 min	Group (HIE_mild_ vs. HIE_cooled_)	−7.81816 (−11.43931, −4.19699)	<0.001 *
	Time	−0.00206 (−0.03730, 0.03318)	0.909
	Group × Time	−0.00630 (−0.05579, 0.04318)	0.803
20 min	Group (HIE_mild_ vs. HIE_cooled_)	−8.29617 (−12.21319, −4.37910)	<0.001 *
	Time	0.00953 (−0.07861, 0.09766)	0.833
	Group × Time	−0.02562 (−0.14949, 0.09826)	0.686
30 min	Group (HIE_mild_ vs. HIE_cooled_)	−8.15252 (−12.17035, −4.13469)	<0.001 *
	Time	0.06239 (−0.07834, 0.20311])	0.386
	Group × Time	−0.10171 (−0.29620, 0.09278)	0.307
60 min	Group (HIE_mild_ vs. HIE_cooled_)	−8.59229 (−12.89037, −4.29420)	<0.001 *
	Time	0.17269 (−0.16906, 0.51445)	0.325
	Group × Time	−0.27497 (−0.74849, 0.19853)	0.258
120 min	Group (HIE_mild_ vs. HIE_cooled_)	−9.24454 (−14.32783, −4.16125)	0.001*
	Time	0.27341 (−0.91995, 1.46678)	0.652
	Group × Time	0.08464 (−1.569787, 1.73906)	0.920

* statistical significance *p* < 0.01.

## Data Availability

The data presented in this study are available on request from the corresponding author since we have not setup a public archive platform for data sharing.

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
