# Peer review of "Feasibility of EEG Phase-Amplitude Coupling to Stratify Encephalopathy Severity in Neonatal HIE Using Short Time Window"

_brainsci, 2022, doi:10.3390/brainsci12070854_

Round 1

Reviewer 2 Report

The manuscript entitled “Feasibility of EEG Phase-Amplitude Coupling to Stratify Encephalopathy Severity in Neonatal HIE using Short Time Window” by Wang et al, an interesting attempt to use a biomarker obtained from EEG to distinguish among different degrees of hypoxic-ischemic encephalopathy (HIE) in newborns. Obviously, the rapid and exact diagnosis of is extremely important to start therapeutic measures as soon as possible to minimize the morbimortality. Therefore, every development to help clinicians will be welcome. However, there are several flaws in the manuscript that must be changed before publication. In my experience, the manuscript has resulted difficult to follow the logic and there are a lot of information that authors suppose the readers have. But I think that a manuscript must be auto-evident and contain most of the information needed for an individualized reader can decide about its quality and even reproduce the methods.

1.- Please, enlisted briefly the most relevant signs of encephalopathy and a sentinel perinatal event.

2.- The authors classify newbors in two groups: fetal acidosis with no encephalopathy (control) and those with moderate to severe HIE. Therefore, what the authors are doing, in fact, is characterizing the values of tPACm in both groups previously defined, not classifying unknown newborns in HIE vs non-HIE. Besides, this is a dichotomic variable, and, at the best, authors can say that a given newborn has or not HIE, but not stratify the severity, as the title, abstract and conclusions indicate. So, it should be clearly stated that the method described can distinguish among HIE and non-HIE but not stratify the severity of HIE.

3.- Are the analysis directly performed onto the EEG recordings or do they previously convert to ASCII/EDF and export it?.

4.- Please, clarify what are the EEG signals with other physiological parameters and why do you need another device to record.

5.- Statistics is poorly explained. Do you assessed normality for distributions?, e.g. for the assumption that t-test can be used for two samples like these.

6.- I’m conscious that authors refer to a previous work for explanation of tPACm, but I think that an enough detailed description must be offered.

7.- Titles from references 4 and 5 seems the same and also the journal. Are these mistakes?.

8.- Please, explain briefly but in a suffice degree what are mixed-effects models. They seem a multivariate linear function, but no references are offered.

9.- Table 1. Why to indicate the umbilical cord prolapse if there are 0 patients. Indicate what is the meaning for the acronym DOL.

10.- I guess that figure 2 computes the results from the whole group of HIE and non-HIE. Firstly, where is indicated at text that non-HIE are HIE mild? This classification is indicated, for example, at table 1. Therefore, it’s quite puzzling because No HIE is different from mild HIE. Secondly, if the results are obtained from a group, you must indicate error measurements.

Reviewer 3 Report

I appreciate the authors presenting this clinical application useful research article. The novelty is enough and the results are rich.  My comments are as follows

1.  Introduction:  please provide the potential biomarker for HIE at present 

2. Methods:

   2.1.  a. please provide the inclusion and exclusion criteria for this study

           b. Why choice fetal acidosis with no encephalopathy as control?

               please  clarify

           c. When  and where to start the recording? 

Round 2

Reviewer 2 Report

I'm satisfied with answers and I have no more questions.

Author Response

Thank you very much.